# A Lean and Performant Hierarchical Model for Human Activity Recognition Using Body-Mounted Sensors

**DOI:** 10.3390/s20113090

**Published:** 2020-05-29

**Authors:** Isaac Debache, Lorène Jeantet, Damien Chevallier, Audrey Bergouignan, Cédric Sueur

**Affiliations:** 1Institut Pluridisciplinaire Hubert Curien (IPHC) UMR 7178 Centre National de la Recherche Scientifique (CNRS), Université de Strasbourg, 67000 Strasbourg, France; lorene.jeantet@iphc.cnrs.fr (L.J.); damien.chevallier@iphc.cnrs.fr (D.C.); audrey.bergouignan@iphc.cnrs.fr (A.B.); cedric.sueur@iphc.cnrs.fr (C.S.); 2Division of Endocrinology, Metabolism, and Diabetes and Anschutz Health and Wellness Center, University of Colorado, School of Medicine, Aurora, CO 80045, USA; 3Institut Universitaire de France, Saint-Michel 103, 75005 Paris, France

**Keywords:** accelerometers, sensors, human activity recognition, machine learning

## Abstract

Here we propose a new machine learning algorithm for classification of human activities by means of accelerometer and gyroscope signals. Based on a novel hierarchical system of logistic regression classifiers and a relatively small set of features extracted from the filtered signals, the proposed algorithm outperformed previous work on the DaLiAc (Daily Life Activity) and mHealth datasets. The algorithm also represents a significant improvement in terms of computational costs and requires no feature selection and hyper-parameter tuning. The algorithm still showed a robust performance with only two (ankle and wrist) out of the four devices (chest, wrist, hip and ankle) placed on the body (96.8% vs. 97.3% mean accuracy for the DaLiAc dataset). The present work shows that low-complexity models can compete with heavy, inefficient models in classification of advanced activities when designed with a careful upstream inspection of the data.

## 1. Introduction

Physical activity monitoring with wearable sensors has various scientific, medical and industrial applications, such as physical activity epidemiology [1], fall detection in the elderly population [2] and for smartwatch applications [3]. Among the existing sensors, accelerometers (sometimes coupled with gyroscopes [4]) are regularly used for activity monitoring, mainly because of their relatively high accuracy, low price and small size [5,6]. Methods for human activity recognition (HAR) using wearable motion sensors were thoroughly investigated and reported in the scientific literature, and a large number of studies demonstrated their ability to predict activity with a high level of accuracy [7,8].

Despite these advances in the field, studies in physical activity epidemiology have mostly used opaque, proprietary algorithms [9,10,11], hence limiting comparability between studies and innovation in the spectrum of activities studied. This situation is probably due to the complexity of the algorithms proposed in the literature, which have grown long and difficult to implement as the HAR tasks became more challenging. Thus, there is a need for a simple yet performant algorithm that scientists could easily implement when analyzing accelerometer data. 

Existing transparent HAR methods usually rely on supervised machine learning models to map between motion signals and activities. All methods rely on the assumptions that different physical activities are reflected by different, characteristic signals and that it should be possible to discriminate between activities with appropriate, meaningful features extracted from the signal [8,12]. HAR models can be divided into two main families: classical machine learning models and neural networks [13]. In the classical approach, activities are discriminated by means of handcrafted features extracted from segments of the signal in the time and frequency domains (e.g., mean, standard deviation or maximum frequency) [8,12]. Such features have proved useful in discriminating activities in various models, such as tree-based models, support vector machines (SVM), logistic regression (LR), k-nearest-neighbors (KNN), naïve Bayes classifiers and hidden Markov models (HMM) [7,12]. In contrast, neural networks can be fed directly with the raw signal and are automatically tuned in order to detect discriminative features [13,14]. Neural networks have been proposed in different variants, such as convolutional neural networks (CNN) and recurrent networks [14]. 

The automatic feature detection of neural network models makes them capable of detecting very complex, highly discriminative features and patterns in the data [13]. CNN drawing upon advances in computer vision have recently proved powerful in HAR and outperformed classical machine learning models (e.g., [15,16]). Although very performant, deep learning models are very slow to train, and finding the optimal architecture for the task at hand is most often a tedious process [14]. The effectiveness of automatic feature learning comes, thus, at a high computational price, which makes it often more efficient to rely on human domain knowledge for feature extraction [13]. Furthermore, the long process of model selection makes the final model hardly generalizable to similar but different tasks [14,17].

Classical supervised machine learning methods, in contrast, are easier to train but their shallow learning can make them less performant in difficult classification tasks [13]. To make up for these deficiencies, researchers using classical models must handcraft a very large number of increasingly complex features, sometimes amounting to several thousand [8,18]. Because too many features can impair the performances of the models and make training computationally impractical, researchers must engage in a process of feature selection in order to form a small subset of highly informative features, which are subsequently fed into the classification models [19]. This process of feature selection can be in itself complex [18], resulting in computationally expensive, inefficient and sometimes unclear classification algorithms. 

Several studies demonstrated the usefulness of a hierarchical classification system for HAR with increasing accuracy while keeping the algorithm reasonably simple [20,21,22]. This system consists of assigning precise target classes to samples in two steps. In the first step, a base classifier discriminates between meta-classes regrouping several similar target classes. In the second step, classifiers specific to each meta-class discriminate between the final target classes. With a strong base-level classifier, such systems can manually prevent potential misclassification [21] and combine different classifiers for different tasks, each “specializing” in a different problem solving task [20]. Finally, a hierarchical system provides an interesting insight into the performance of the algorithm solving a basic classification, which can represent an objective per se. 

The goal of this article is to propose a high-performance, fast and easy-to-implement algorithm for HAR, which could compete with state-of-the-art complex algorithms, including those based on neural network models. The proposed algorithm relies on a novel hierarchical system and a relatively small set of highly informative features extracted from the filtered signals, and was evaluated against the public Daily Living Activity (DaLiAc) dataset presented below [20]. Because this algorithm was specifically designed for the classification task of the DaLiAc dataset, we further assessed its generalizability by testing it against another dataset, the mHealth dataset [23]. Finally, given that many popular activity monitors (e.g., ActiGraph or ActivPal in health studies) are not equipped with gyroscopes, we assessed the usefulness of adding gyroscopes to the accelerometers by comparing the performance of the algorithm when gyroscope data were included and when they were not.

## 2. Materials and Methods

### 2.1. The DaiLAc Dataset

The DaLiAc (Daily Living Activity) dataset consists of the signals of accelerometers and gyroscopes placed on the chests, wrists, hips and ankles of 19 adults performing thirteen daily activities in semi-controlled conditions. The activities include a wide range of simple and complex activities: lying, sitting, standing, dish washing, vacuum-cleaning, sweeping, walking, running, ascending stairs, descending stairs, bicycling with a resistance of 50 Watts, bicycling with a resistance of 100 Watts and rope jumping. Details about the subjects and the experimental designs can be found elsewhere [20]. 

### 2.2. Processing

Acceleration signals are known to be composed of a dynamic component (acceleration of the body) and a gravitational one. As a consequence, some authors suggested applying a low-pass filter to the acceleration signal in order to isolate the gravitational component and infer the inclination of the device in space [8,24]. Using a Butterworth filter (first order, with a threshold of 2 Hz), we separated the accelerometer signals into dynamic and gravitational components (AC and DC components, respectively). Unlike the widespread approach, we treated raw acceleration, AC and DC components as three separate signals all along the feature extraction process. AC and DC components reflect two different aspects of physical activity, orientation and motion, and as such should be treated as two independent signals. For instance, periodicity metrics extracted for the signals can be different, but equally interesting, when looking at orientation and motion over time. Thus, we ended up, for each sensor, with the following time-series: three total acceleration signals (along each axis), three AC, three DC and three gyroscope signals. All signals were downsampled to 51.2 Hz (we sampled every fourth datapoint from the original data) and normalized. 

All signals were segmented along the time axis into windows of five seconds with a 50% overlap, as done by other authors [25], in order to make evaluation comparable with other algorithms tested on the same data [15].

### 2.3. Feature Extraction

We define as ***x*** the signals (raw accelerometer and gyroscope data, AC and DC) over an *N*-length window (here, we used 5-s windows and a sampling frequency of 51.2 Hz, hence *N* = 256). For each windowed signal ***x***, the following statistics were computed in the time-domain:-Mean, standard deviation, skewness and kurtosis;-The following percentiles: [0, 5, 10, 20, 30, 40, 50, 60, 70, 80, 90, 95, 100];-Range: max(***x***) − min(***x***);-RMS: $\sqrt{\frac{1}{n}\;sum\left(x^{2}\right)}$;-Zero-crossing: the number of times the signal crossed the mean.

To the mean-subtracted signal $x^{\prime }=x-\bar{x}$, we applied the Fourier transformation. We define an amplitude vector $\hat{x}$ as the absolute values of the Fourier transform: (1)$$\hat{x}=\{\left|\hat{f}\left(\xi \right)\right|\;|\;\xi \;\in \left[0\;,+\frac{N}{2}\right]\}$$

The following frequency domain features were computed for all vectors $\hat{x}$:
-Energy: $E=sum\;\left({\hat{x}}^{2}\right)$;-Entropy: $H=-\frac{p\cdot \log\left(p\right)}{\log_{2}\left(\frac{N}{2}\right)}$, where $p=\frac{\hat{x}}{sum\left(\hat{x}\right)}$;-Centroid: c=ξ·p, where $\xi =\{\xi |\;\xi \;\in \left[0\;,+\frac{N}{2}\right]\}$;-Bandwidth: b=δ·p, where δ=ξ−c;-Maximum frequency: $argmax\left(\hat{f}\left(\xi \right)\right)$.

### 2.4. Classification

Classification was done using a two-level hierarchical system, as illustrated in Figure 1. For all classification tasks in the system, the following classifiers were tested: LR (with a L2 regularization and a penalty coefficient equal to one); KNN with k = 5; gradient boosting (GB) (500 estimators, selecting 10 features at a time); and SVM. For additional comparability, a convolutional network was also tested (architecture in Figure 2) taking as input the four signals (AC, DC, accelerometer and gyroscope) and their Fourier transform. Classification was done using all 15 possible combinations of device locations on the subjects’ body (e.g., ankle, ankle + chest and ankle + chest + wrist).

We used Python’s Scikit-learn [26] and Tensorflow [27] libraries for the analysis, and unless otherwise specified, their default parameters. The Python scripts of the project are available on the Github repository (see Appendix A).

### 2.5. Evaluation Method

In order to evaluate the performances of the proposed models, a leave-one-subject-out procedure was followed: models were tested against data from one subject after being trained on all others, for each subject of the 19 subjects in the dataset. This procedure was adopted by the first study on the dataset and followed by several subsequent studies (Table 1). Reserving a fraction of each subject’s data for testing instead a fraction of the subjects themselves can result in an upward bias of the estimate of the performance metric, since models learn the patterns that are specific to the subjects and can better classify them during testing. Moreover, averaging scores of all iterations in a leave-one-subject-out procedure is preferable to a single hold-out test on a several subjects, as it reduces bias in the accuracy estimator, especially in small datasets [20].

For all models, we reported the mean and standard deviation of the accuracy (rate of correctly classified samples) for the 19 leave-one-subject-out rounds. To present a complete picture, for models based on the four devices, we also presented the confusion matrix, and the f-score, which is the harmonic mean of precision (true positives/(true positives + false positives)) and recall (true positives/(true positives + false negatives)).

### 2.6. Generalization on the mHealth Dataset

The algorithm presented in this article was designed to address the specific classification task of the DaLiAc dataset. It was therefore deemed desirable to validate this algorithm on other data, collected in different conditions and presenting a different classification task. To do so, we used the algorithm on the mHealth dataset [23] that contains labelled body-worn accelerometer, gyroscope and magnetometer signals collected while subjects were performing different activities. The accelerometer, gyroscope and magnetometer sensors were placed on the lower arm and the ankle. In addition, a device placed on the chest recorded accelerometer data only. Data for the activities were collected in an out-of-the lab environment with no constraints on the way activities must be executed; subjects were asked to try their best when executing them. The activities were the following: standing still, sitting and relaxing, lying down, walking, climbing stairs, bending the waist forward, frontal elevation of arms, bending the knees (crouching), cycling, jogging, running and jumping forwards and backwards. We trained and tested the data using the exact same algorithm, hyper-parameters and validation procedure as those presented here for the DaLiAc dataset. We used a flat classification, since classes seemed clearly distinct from each other. 

## 3. Results

### 3.1. Results for the DaLiAc Dataset

For the five classification models (LR, GB, KNN, SVM and CNN), accuracy is reported for each combination of devices and for each task in the hierarchical system (Table 2, and in Appendix A). Overall classification accuracy was highest for LR (based on data from all four devices) with 97.30% accuracy, followed by GB (all devices) with 96.94%, SVM (all devices) with 96.84%, CNN (three devices, ankle, chest and wrist) with 95.42% and KNN (three devices, ankle, chest and wrist) with 91.82%. When looking at sub-tasks in the hierarchical classification system, GB is very slightly better than LR in the base-level classification (99.23% vs. 99.21%). GB outperformed LR also in distinguishing between standing and washing dishes (97.40% vs. 97.06%) and between walking, ascending and descending stairs (99.08% vs. 98.72%). When we combined the best classifiers for all sub-tasks, overall mean accuracy rose by 0.04%. As this improvement remains very marginal, we refer to the system based exclusively on LR as the best algorithm. The confusion matrix for the final classification with LR is shown in Table 3.

The training time varied significantly across the models studied. Using Google Colab (with GPU accelerator) and the parameters mentioned above, training and predicting data following the leave-one-out procedure (i.e., 19 times) for the DaLiAc dataset lasted 4.5 min with LR and KNN, 7.2 min for SVM, 10.7 min for GB and over half an hour for CNN (Table 2). The entire feature extraction phase for the 19 subjects (over six hours of observations in total) took about 30 s.

Regarding the locations of the devices on the body, the best choices of one, two and three locations out of the four studied were chest (93.39% with SVM), ankle + wrist (96.81% with LR) and ankle + wrist + chest (97.06% with LR), respectively (Table 2). Table 4 shows a comparison of the classification accuracies based on both accelerometers and gyroscopes with those obtained with accelerometers only. The loss in mean accuracy was relatively small when leaving out gyroscopes (−0.4%, −0.4%, −0.1% and −2.5% for the best four, three, two and one locations using LR, respectively).

### 3.2. Results for the mHealth Dataset

The ability of our algorithm to generalize was further validated on the mHealth dataset, using all accelerometer, gyroscope and magnetometer signals. We obtained very good average scores on the mHealth dataset with GB (98.7% ± 2.6%), LR (98.2% ± 2.7%) and SVM (97.2% ± 3.7%), but less good ones with KNN (92.7% ± 4.3%) and CNN (87.7% ± 7.5%).

## 4. Discussion

Compared with previous works tested on the DaLiAc data set, the proposed algorithm, based on careful handcrafted features extracted from the signals, represents a threefold improvement. First, the proposed algorithm performs better than major works tested against the DaLiAc dataset (97.30% accuracy with LR versus 96.40% for the best model so far with CNN [15]) (see Table 1). Likewise, our algorithm with GB and LR yielded less than 2% classification error on the mHealth dataset. By comparison, Jordano et al. [31] identified seven studies evaluated against the mHealth dataset, and when applying the same leave-one-subject-out procedure, the accuracy for the best algorithm was 94.66%. Zdravevski et al. [18] using a hold-out dataset for testing (subjects 7–10) reached 99.8% accuracy. By applying the same procedure and the same windowing strategy, we reached an accuracy of 99.7% with our algorithm (LR).

Second, compared to state-of-the-art CNN, the proposed algorithm performed best with fast-training models, such as logistic regression (32 min for the former versus 4.5 min for the latter).

Third, these superior results were obtained with simple and robust tools in machine learning that do not require preliminary hyper-parameter optimization and feature selection, such as LR. In fact, hyper-parameters optimization of classifiers (most notably neural networks) and feature selection can be a daunting, time-consuming task, and was shown to lead to over-fitting and poor generalization [32]. This was corroborated by the validation of the algorithm against the mHealth dataset. Simple classifiers based on handcrafted features, which required no or little hyper-parameter tuning, generalized very well on a new dataset, while CNN, which performed well on DaLiAc, for which it was tuned, yielded poor results on mHealth.

It is difficult to fully explain how our algorithm outperformed previous algorithms using classical machine learning classifiers by around 4%, as authors do not always specify all the decisions that they make during data processing before reaching the results. Using the DaLiAc dataset, we undertook a few steps to identify the innovations that made our algorithm more accurate. First, running our algorithm with a flat classification system instead of the hierarchical system proposed here resulted in 1.81% decrease in mean accuracy. Second, by extracting features performed on the acceleration signal only, without including the AC and DC components as we did, the decrease in accuracy amounted to 2.63%. The additional 1.27% difference with the two best-performing algorithms using classical methods by Chen [28] and by Zdravevsky [18] can be attributed to a good trade-off between the number of features and their informativeness. In fact, the former study omitted very important features (i.e., no frequency domain features were extracted), while the latter may have had too many of them (4871 before selection). 

Large-scale past public health studies in activity monitoring, such as NHANES [1], have relied only on accelerometer sensors to derive activities. Yet, many of the state-of-the-art algorithms have been developed for a combination of accelerometer and gyroscope data. We have shown here that with our algorithm, the decrease in accuracy following the removal of gyroscope signals was marginal. This will help designers of future studies make an informed decision about the trade-off between cost and accuracy.

Despite this promising improvement, two caveats need to be highlighted. The first caveat relates to the nature of our data. HAR algorithms are tested against clean data of activities performed in a characteristic manner as part of a relatively structured protocol. Realistic data, however, can contain fewer characteristic activities (e.g., slouching) which represent a greater challenge to classify. To that extent, very recent attempts to create benchmark activity datasets simulating real conditions [33] are an important development in the field and new algorithms should preferably be assessed using these data. In addition, people in real conditions tend to switch rapidly between activities. Consequently, windows of five seconds are probably too long to capture a single activity. A possible solution would be to view sets of activities that are often performed together (e.g., standing and walking around) as activities per se. Another solution is to consider smaller windows, for instance, of one second. Smaller windows are known to be less good when aiming to capture cyclical activities [25] and can result in a decrease in total accuracy and longer training. In fact, running our algorithm on one-second windows resulted in a drop of 2.9% and lasted almost five times as long as with the five-second windows commonly used (data not shown). Limiting this loss in accuracy by applying dynamic windowing methods [25,34] is an interesting direction for future development. 

A second caveat pertains to the ranking of the models tested in this study. A better choice of the hyper-parameters of the powerful SVM, GB or CNN models could have resulted in another ranking. Our points are to emphasize that a simple approach based on domain knowledge can result in a fast, robust and performant model; and that issues of generalizability and tedious processes of model selection must be acknowledged in the evaluation of a new algorithm.

## 5. Conclusions

In this paper, we propose a novel algorithm for HAR from motion signals (accelerometers and gyroscopes), which significantly improves upon previous work in terms of computational expenses, inferential robustness and classification accuracy. Using a hierarchical classification system with LR, and a relatively small set of features extracted not only from the acceleration signal, but also from low-pass filtered and high-pass filtered signals, proved highly useful in solving our classification task. From a practical perspective, we showed that two devices placed on the wrist and the ankles resulted in an accuracy that is practically as good as with two additional accelerometers on the chest and the hip, and that using the method proposed here, the additional information brought by the gyroscope was marginal. 

Future research should focus on data that better simulate real life conditions, with their swift transitions between activities and less characteristic behaviors. New, simple models should be developed to better adapt to these conditions, while relying, as much as possible, on domain knowledge. 

## Figures and Tables

**Figure 1 sensors-20-03090-f001:**
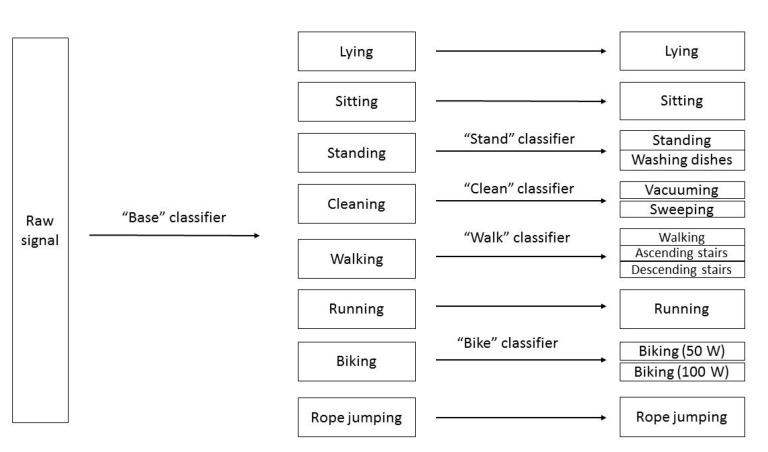
Illustration of our hierarchical classification system.

**Figure 2 sensors-20-03090-f002:**
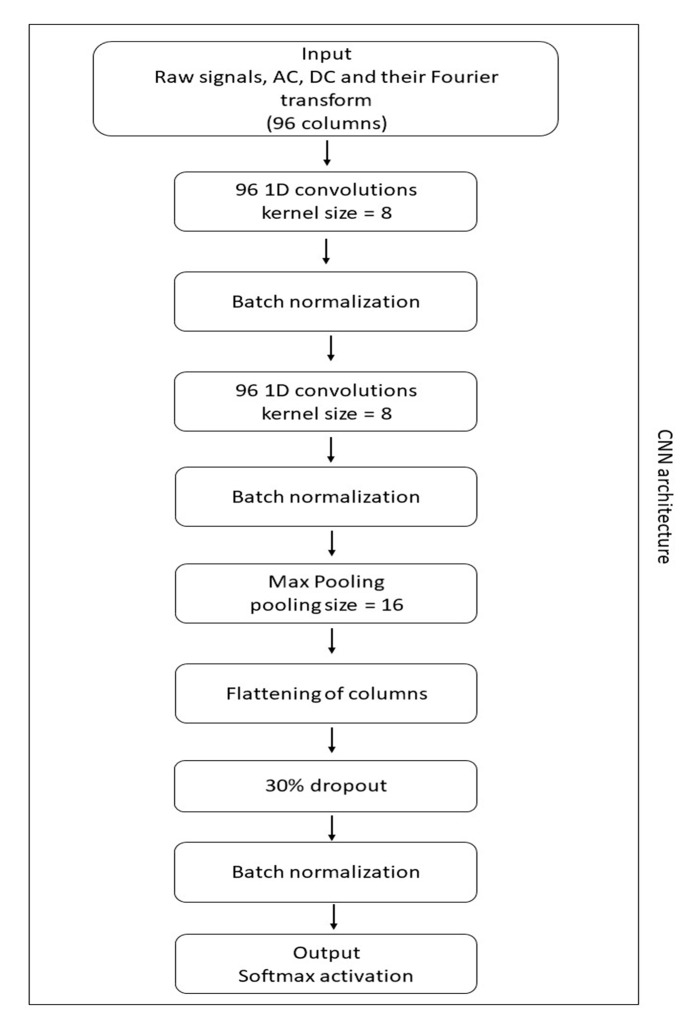
The architecture of the convolutional neural network tested here. Except for the output, all layers were activated with the *RELU* function.

**Table 1 sensors-20-03090-t001:** Overview of previous algorithms applied to the DaLiAc dataset (with testing on unseen subjects).

Authors	Year	Classifiers	Mean Accuracy Score (%)	Remark
Leutheuser et al. [20]	2013	SVM, AdaBoost, KNN	89.6	Reference paper
Chen et al. [28]	2016	SVM	93.4	
Nazabal et al. [29]	2016	HMM	95.8	Merged the two bicycle activities
Zdravevski et al. [18]	2017	SVM	93.4	
Hur et al. [15]	2018	CNN	96.4	
Jurca et al. [30]	2018	LSTM	87.2	
Huynh-The et al. [16]	2019	CNN	95.7	
Proposed algorithm	2020	LR	97.3	

SVM = support vector machine; KNN = k nearest neighbors; HMM = hidden Markov model; CNN = convolutional neural network; LSTM = long short time memory; LR = logistic regression.

**Table 2 sensors-20-03090-t002:** Best mean (maximum) and standard deviation (minimum) of accuracy score by classification task and classifier.

Task →	Base		Stand/Washing Dishes		Vacuum/Sweep		Walk/Ascending Stairs/Descending Stairs		Bike 50 Watt/ Bike 100 Watt		Overall		Execution Time
↓ Classifiers	Mean	sd	Mean	sd	Mean	sd	Mean	sd	Mean	sd	Mean	sd	
SVM	0.9911	0.0076	0.9716	0.0365	0.9397	0.0521	0.9872	0.0076	0.9495	0.0577	0.9684	0.0166	7.2 min
Best sensor combination	ACHW	AH	ACHW	AHW	ACHW	HW	A	A	ACH	C	ACHW	AHW	
CNN	0.9896	0.0093	0.965	0.0498	0.9364	0.0607	0.9799	0.0168	0.9259	0.0577	0.9542	0.022	32.0 min
Best sensor combination	ACW	ACW	AW	A	ACW	ACW	ACH	ACHW	AHW	ACH	ACW	ACW	
KNN	0.984	0.0128	0.9336	0.0742	0.8642	0.0633	0.9873	0.0085	0.8042	0.0754	0.9182	0.0233	4.5 min
Best sensor combination	ACW	AW	ACHW	ACHW	ACW	ACW	AC	AC	AC	ACH	ACW	ACW	
GB	0.9923	0.0057	0.974	0.0313	0.9292	0.0487	0.9908	0.0063	0.9408	0.0546	0.9694	0.0188	10.7 min
Best sensor combination	ACH	AHW	ACHW	ACHW	ACHW	AHW	ACH	ACH	ACW	CHW	ACHW	ACHW	
LR	0.9921	0.0069	0.9706	0.0354	0.9444	0.0453	0.9872	0.0099	0.9547	0.0493	0.973	0.0135	4.5 min
Best sensor combination	AHW	AW	ACW	AHW	ACW	AHW	AC	A	ACHW	AW	ACHW	AW	

Legend: SD = standard deviation, A = ankle, C = chest, H = hip, W = wrist.

**Table 3 sensors-20-03090-t003:** Aggregated confusion matrix for all leave-one-subject-out rounds (logistic regression). Class-specific precision, recall and f-score (β = 1) are reported for each class of the DaLiAc dataset. Values in bold (diagonal) represent correct predictions.

	Sit	Lie	Stand	Wash	Vacuum	Sweep	Walk	Stairs-Up	Stairs-Down	Run	Bike 50W	Bike 100W	Jump
**sit**	**430**	0	17	3	0	0	0	0	0	0	0	0	0
**lie**	1	**455**	0	0	0	0	0	0	0	0	0	0	0
**stand**	2	0	**442**	8	0	0	1	0	0	0	0	0	0
**wash**	0	0	2	**924**	7	4	0	0	0	0	0	0	0
**vacuum**	0	0	0	7	**422**	25	0	0	0	0	0	0	0
**sweep**	0	0	6	4	23	**704**	4	2	0	0	0	0	0
**walk**	0	0	3	1	4	5	**2010**	11	6	1	0	0	0
**stairsup**	0	0	0	0	0	1	6	**312**	1	0	0	0	0
**stairsdown**	0	0	0	0	0	0	5	2	**266**	0	0	0	0
**run**	0	0	0	0	0	0	0	0	0	**910**	1	0	0
**bike 50W**	0	0	0	0	0	0	0	0	0	0	**877**	46	0
**bike 100W**	0	0	0	0	0	0	0	0	0	0	37	**883**	2
**jump**	0	0	0	0	0	0	0	0	0	0	0	0	**243**
**precision**	0.993	1.00	0.940	0.976	0.926	0.953	0.992	0.954	0.974	0.999	0.959	0.950	0.992
**recall**	0.956	0.998	0.976	0.986	0.930	0.948	0.985	0.975	0.974	0.999	0.950	0.958	1.000
**f_score**	0.974	0.999	0.958	0.981	0.927	0.950	0.989	0.964	0.974	0.999	0.954	0.955	0.996

**Table 4 sensors-20-03090-t004:** Comparison of classification accuracy on the DaLiAc dataset with versus without gyroscope data for all combinations of devices.

	Accelerometer/Gyroscope		Accelerometer Only		Mean Difference
	Mean	sd	Mean	sd	
**ankle**	0.920	0.03	0.921	0.02	0.0010
**chest**	0.926	0.03	0.901	0.03	0.0250
**hip**	0.894	0.04	0.867	0.05	0.0270
**wrist**	0.867	0.5	0.809	0.05	0.0580
**ankle|chest**	0.959	0.02	0.954	0.02	0.0050
**ankle|hip**	0.943	0.03	0.941	0.02	0.0020
**ankle|wrist**	0.968	0.01	0.958	0.01	0.0100
**chest|hip**	0.943	0.03	0.93	0.03	0.0130
**chest|wrist**	0.954	0.02	0.934	0.03	0.0200
**hip|wrist**	0.945	0.03	0.926	0.03	0.0190
**ankle|chest|hip**	0.960	0.02	0.956	0.02	0.0040
**ankle|chest|wrist**	0.970	0.02	0.966	0.02	0.0040
**ankle|hip|wrist**	0.968	0.01	0.964	0.01	0.0040
**chest|hip|wrist**	0.962	0.02	0.949	0.02	0.0130
**ankle|chest|hip|wrist**	0.973	0.02	0.969	0.02	0.0040

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
