# Peer review of "A Lean and Performant Hierarchical Model for Human Activity Recognition Using Body-Mounted Sensors"

_sensors, 2020, doi:10.3390/s20113090_

Round 1
Reviewer 1 Report
This paper presents a novel algorithm to determine activity based upon motion signals from accelerometers and gyroscopes. The work shows improvements over past work in terms of computational expenses and classification accuracy.
The authors used a hierarchical classification system with logistic regression. A range of locations and combinations of locations, were tested and quantitative evaluations offered. 5 classification models (logistic regression, gradient boosting, KNN, SVM and CNN) were tested and reported upon. The authors consider that their solution offers a threefold improvement compared with previous work.
Notably, using 2 devices placed on the wrist and the ankle gave comparable accuracy with another 2 additional devices on the chest and hip. This offers a practical reduction in the number of required devices (and thus cost and complexity).
Some issues need addressed. Figure 2 is unclear in its intention, should there be arrows, if it is an architecture then I recommend a box around it to show it is part of the same system, etc (whatever you feel is appropriate to make the figure more usable please implement that).
Some formatting issues exist, e.g. Table 1 heading is not on the same page as Table 1.
Overall, this is a well written paper with a significant interest to readers. The outcomes are very usable and can be utilised by other researchers and developers to create ADL identification systems. This is particularly useful for the elderly living on their own as monitoring their ADL is a direct indicator to their physical and mental well-being
Author Response
COMMENT: This paper presents a novel algorithm to determine activity based upon motion signals from accelerometers and gyroscopes. The work shows improvements over past work in terms of computational expenses and classification accuracy.
The authors used a hierarchical classification system with logistic regression. A range of locations and combinations of locations, were tested and quantitative evaluations offered. 5 classification models (logistic regression, gradient boosting, KNN, SVM and CNN) were tested and reported upon. The authors consider that their solution offers a threefold improvement compared with previous work.
Notably, using 2 devices placed on the wrist and the ankle gave comparable accuracy with another 2 additional devices on the chest and hip. This offers a practical reduction in the number of required devices (and thus cost and complexity).
Some issues need addressed. Figure 2 is unclear in its intention, should there be arrows, if it is an architecture then I recommend a box around it to show it is part of the same system, etc (whatever you feel is appropriate to make the figure more usable please implement that).
ANSWER: We have made the legend in Table 2 more explicit and added a box and arrow to the diagram.
COMMENT: Some formatting issues exist, e.g. Table 1 heading is not on the same page as Table 1.
ANSWER: The issue has been addressed
COMMENT: Overall, this is a well written paper with a significant interest to readers. The outcomes are very usable and can be utilised by other researchers and developers to create ADL identification systems. This is particularly useful for the elderly living on their own as monitoring their ADL is a direct indicator to their physical and mental well-being
ANSWER: We thank the reviewer for this positive comment.
Reviewer 2 Report
Dear authors,
The paper introduces a method to recognize human activities. The task is enjoyable, and the paper presents the problem correctly, and the conclusions are adequate and supported by the experiments.
The paper has to be completed by providing the input data of your algorithm, the script configuration file you used, and a script to elaborate the input data. The technical questions are strictly related to your work.
Best regards.
Author Response
COMMENT: Dear authors,
The paper introduces a method to recognize human activities. The task is enjoyable, and the paper presents the problem correctly, and the conclusions are adequate and supported by the experiments.
The paper has to be completed by providing the input data of your algorithm, the script configuration file you used, and a script to elaborate the input data. The technical questions are strictly related to your work.
ANSWER: The datasets have been uploaded to Github. The script was uploaded, thoroughly commented and illustrated, as a Jupyter notebook, providing the reader with a clear end-to-end overview of our work, including the processing of the input data.
Reviewer 3 Report
The problem posed by the article is that state of the art methods in HAR are often obscure and difficult to generalise, and the authors propose a hierarchical logistic regression classifier with the aim to demonstrate that, with the help of domain knowledge, it is possible to derive a faster and more general classification algorithm which can nevertheless compete with the state of the art. However, the paper does not give sufficient evidence to support this position.
The classifier is compared against state of the art methods on the daily living activities (daliac) dataset, and performs better than the state of the art. However, the claim that this classifier is more easily generalisable is not supported, since only one dataset was evaluated - my suggestion is to include comparisons with state of the art methods on other datasets with little or no hyperparameter tuning to demonstrate the superiority of the hierarchical LR classifier. In addition, the dataset chosen (daliac) is perhaps not the most impressive since activities tend to be periodic and distinct, and therefore perhaps easier to classify with a simpler method. The real test of this algorithm would be does it still perform well enough on more complex datasets, and if not is the increase in flexibility and ease of implementation enough to outweigh the accuracy drop.
In the discussion it is said that other methods (GB, CNN) could have shown better results with more hyperparameter tuning. This is fine if the rest of the paper highlights the generalisability and performance improvements of the LR method - for example the above and some graphs or tables showing the processing time / memory usage of the proposed classifier compared to some state of the art methods. With those additions I think the paper would have a very strong message.
Outside of this issue of message the paper is well written, some additional points on presentation and style below.
The authors should change the equations in section 2.3 to vectorial format.
The authors should explain a bit more the choice of leave-one-out evaluation - why does everyone use it and what are the implications?
The table that is there is confusing and the tables that are not there contain useful information - I would say to include a combined version of tables 2 and 3 and have the full versions on the github. For example show the results for the best combination of sensors for each algorithm and say what these combinations are in a caption, then have the other combinations in the supplementary material. This makes it more clear in the paper why logistic regression was chosen over the other methods.
It is not clear what value table 5 has or why it is included - it is interesting that removing the gyroscope data gives a small drop in accuracy but not clear how that relates to the message of the paper. The authors either clarify in the discussion or remove it.
Tables 2 and 4 should be aligned together properly.
Author Response
REVIEWER 3
COMMENT: The problem posed by the article is that state of the art methods in HAR are often obscure and difficult to generalise, and the authors propose a hierarchical logistic regression classifier with the aim to demonstrate that, with the help of domain knowledge, it is possible to derive a faster and more general classification algorithm which can nevertheless compete with the state of the art. However, the paper does not give sufficient evidence to support this position.
The classifier is compared against state of the art methods on the daily living activities (daliac) dataset, and performs better than the state of the art. However, the claim that this classifier is more easily generalisable is not supported, since only one dataset was evaluated - my suggestion is to include comparisons with state of the art methods on other datasets with little or no hyperparameter tuning to demonstrate the superiority of the hierarchical LR classifier. In addition, the dataset chosen (daliac) is perhaps not the most impressive since activities tend to be periodic and distinct, and therefore perhaps easier to classify with a simpler method. The real test of this algorithm would be does it still perform well enough on more complex datasets, and if not is the increase in flexibility and ease of implementation enough to outweigh the accuracy drop.
ANWER: We have performed a further validation on the mHealth dataset, using the same models and hyperparameters designed for the DaiLAc dataset in order to assess, as you suggested, the generalizability of our conclusions. The script is available in the Github repository.
We obtained a very good average score with gradient boosting (98.7%±2.6%), logistic regression (98.2%±2.7%), and support vector machine (97.2%±3.7%), but much less good with k-nearest neighbors (92.7%±4.3%) and convolutional neural network (87.7%±7.5%) The good results and the significant drop in the CNN accuracy for the mHealth dataset (88%) compared to the DaiLAc dataset (96%) seem to support our claims about the good performance and generalizability of our algorithm based on simple classifiers in the feature space.
A text regarding this additional evaluation was added in the Introduction (lines 83-85), Material and Methods (lines 170-184), Results (lines 216-220), and Discussion (lines 231-236 and 244-247) sections.
It should be noted that in both datasets, logistic regression and gradient boosting reach about the same level of accuracy. To that extent, one cannot be said to be clearly superior to the other. However, LR is a simpler tool insofar as it requires no or very little hyperparameter tuning and it is faster. Moreover, our main point is to highlight the relevance of simple classifiers based on handcrafted features as a good alternative to neural networks (which emerge as state-of-the-art methods) in terms of accuracy, simplicity of design, training time and -- if we can conclude so from our evaluation against the mHealth dataset -- generalizability. An effort was made to sharpen this message in the discussion (lines 244-247).
Regarding the dataset chosen, the problem raised by the reviewer seems hardly avoidable in labelled data collected following a rigorous scientific protocol. The issue is acknowledged in the discussion and there is no doubt that our algorithm would be somewhat less performant in real-life settings. However, we still observe that our algorithm represents a relative improvement compared to other algorithms tested on the same data. In addition, it should be noted that, in the DaiLAc dataset, some activities such as walking or ascending/descending stairs were not periodic or performed over distinct time slots. Last, it seems that the subjects in the mHealth studies had some extent of freedom in the execution; as the authors of the dataset claim: “The activities were collected in an out-of-lab environment with no constraints on the way these must be executed, with the exception that the subject should try their best when executing them”. Despite this relative freedom, our algorithm performs very well on this dataset as well, which could testify to its ability to fit to data collected in various conditions.
COMMENT: In the discussion it is said that other methods (GB, CNN) could have shown better results with more hyperparameter tuning. This is fine if the rest of the paper highlights the generalisability and performance improvements of the LR method - for example the above and some graphs or tables showing the processing time / memory usage of the proposed classifier compared to some state of the art methods. With those additions I think the paper would have a very strong message.
ANSWER: The processing time for the different models was added to Table 2.
COMMENT: Outside of this issue of message the paper is well written, some additional points on presentation and style below.
The authors should change the equations in section 2.3 to vectorial format.
ANSWER: This has been done, see section 2.3 lines 117-134.
COMMENT: The authors should explain a bit more the choice of leave-one-out evaluation - why does everyone use it and what are the implications?
ANWSER: We have added some explanation in the M&M to justify this evaluation procedure (section 2.5 lines 152-161).
COMMENT: The table that is there is confusing and the tables that are not there contain useful information - I would say to include a combined version of tables 2 and 3 and have the full versions on the github. For example show the results for the best combination of sensors for each algorithm and say what these combinations are in a caption, then have the other combinations in the supplementary material. This makes it more clear in the paper why logistic regression was chosen over the other methods.
ANSWER: We thank the reviewer for this suggestion. We have made changes accordingly. (see Table 2).
COMMENT: It is not clear what value table 5 has or why it is included - it is interesting that removing the gyroscope data gives a small drop in accuracy but not clear how that relates to the message of the paper. The authors either clarify in the discussion or remove it.
ANSWER/ We have added a paragraph in the introduction and in the discussion about the importance of this comparison (section 1, lines 85-88 and discussion, lines 260-265).
COMMENT: Tables 2 and 4 should be aligned together properly.
ANSWER: This has been done too, thank you for your valuable insight
Round 2
Reviewer 2 Report
Dear authors,
You have replied to my questions. The paper is accepted.
Best Regards.
Reviewer 3 Report
I am satisfied that the message of the paper has been clarified significantly from the original version, especially with the inclusion of the mHealth dataset.
There remain some minor formatting issues - page numbers are wrong after page 6 and the tables 4 and 5 should be aligned better.